# Second-dose measles vaccination coverage and associated factors among children aged 19 to 35 months in Debre Birhan city, Ethiopia, 2024: A community-based cross-sectional study

Tebabere Moltot[1]*, Zenebe Abebe Gebreegziabher[2], Agmasie Damtew Walle[2], Mistire Nigussie[3], Micheal Amera Tizazu[2], Addisalem Workie Demsash[2], Eyosias Yeshialem[2]

**1** School of Nursing and Midwifery, Asrat Woldeyes Health Science Campus, Debre Berhan University, Debre Birhan Ethiopia, **2** School of Public Health, Asrat Woldeyes Health Science Campus, Debre Berhan University, Debre Birhan, Ethiopia, **3** School of medicine, Asrat Woldeyes Health Science Campus, Debre Berhan University, Debre Birhan, Ethiopia

* mtebabere@gmail.com

## Abstract

### Background

Achieving high coverage of both measles vaccine doses is essential for herd immunity. Despite high first-dose coverage, measles outbreaks persist in Debre Birhan City, Ethiopia. Therefore, this study aimed to assess second-dose measles vaccine coverage.and its associated factors among children aged 19–35 months in Debre Birhan City, Ethiopia.

### Methods

A community-based cross-sectional study was carried out in Debre Birhan City from May 10–30, 2024. Systematic random sampling method was used to recruit 626 children aged 19–35 months. Data were collected through a pre-tested, interviewer-administered questionnaire using Kobo Toolbox and later analyzed with SPSS-25. Bivariable and multivariable logistic regression analyses were performed, with statistically significant factors identified at a 95% confidence interval and a p-value of less than 0.05 in the multivariable analysis.

### Results

A total of 626 children participated in the study, achieving a 98.7% response rate. The second-dose measles vaccination coverage among children aged 19–35 months in Debre Birhan City was 57.3% (95% CI: 53.5–61.2). Significant factors associated with the second-dose of measles vaccination coverage were: father's education level: primary (AOR: 1.63; 95% CI: 1.17–3.69), secondary (AOR: 2.04; 95% CI: 1.83–7.01),

**Data availability statement:** Yes - all data are fully available without restriction; "All relevant data are within the paper and its supporting Information files.

**Funding:** The author(s) received no specific funding for this work.

**Competing interests:** The authors have declared that no competing interests exist.

**Abbreviations:** ANC, Antenatal care; MCV1, Measles containing vaccine one; MCV2, Measles containing vaccine two; PNC, Post-natal care; TT, Tetanus toxoid.

and college or above (AOR: 3.2; 95% CI: 1.5–6.95). Other significant predictors were the child being vaccinated for any vaccine (AOR: 3.04; 95% CI: 1.33–6.98), having canceled or postponed vaccination schedules (AOR: 4.9; 95% CI: 2.62–9.32), good level of awareness (AOR: 3.3; 95% CI: 2.12–5.17), and high level of perception (AOR: 5.1; 95% CI: 3.16–8.20).

## Conclusion

Second-dose measles vaccination coverage in Debre Birhan City is lower than the national and global target. Key factors influencing coverage include the father's educational level, canceled or postponed vaccination schedules and caregivers' awareness and perception of measles vaccination. Therefore, to improve coverage, it is essential to strengthen health education, streamline vaccination schedules, and enhance communication with parents.

## Introduction

Despite the existence of a safe and affordable vaccine, measles continues to be a major contributor to global morbidity and mortality. According to the latest WHO estimates, in 2023 there were approximately 10.3 million measles cases and 107,500 deaths, primarily among unvaccinated or under-vaccinated children under the age of five. These modelled figures reflect a 20% increase in global measles cases compared to previous years [1,2]. The measles-containing vaccine (MCV) is one of the most cost-effective public health interventions. Globally between 2000 and 2021, an estimated 56 million deaths were prevented by MCV [3]. Unfortunately, 2020 saw the largest increase in unvaccinated children since 2000, reversing progress in measles elimination efforts [2]. Achieving 95% coverage for both the first (MCV1) and second (MCV2) doses is critical to halting measles transmission. However, maintaining high vaccination coverage among children, especially in low-income countries, remains a significant challenge [4].

Low-income countries, where the risk of measles mortality is highest, also have the lowest vaccination rates, at just 66% [5], and have not yet recovered from the setbacks caused by the pandemic [6]. Over half of the 22 million children who missed their first measles vaccine in 2022 come from 10 countries, including Ethiopia, India, Nigeria, and Indonesia [5,7].

Ethiopia is one of the top five countries with the highest number of unimmunized children, with 1.2 million missing the first dose of the measles vaccine in 2018 [8]. In early 2019, the Expanded Program on Immunization (EPI) began administering the MCV2 vaccine at 15 months to ensure adequate protection [9]. A 95% coverage rate for both MCV1 and MCV2 is needed to eliminate measles circulation. However, achieving and maintaining high vaccination rates, particularly among children aged 19–35 months, continues to be a challenge in many communities [4]. By late 2019, only about 9.1% of children had received the MCV2 dose nationally [10]. According to WHO and UNICEF estimates, MCV2 coverage in Ethiopia was 46% in both 2020 and 2021, and 48% in 2022 [11].

In late 2023 and early 2024, Debre Birhan City, Ethiopia, experienced two measles outbreaks, despite high reported MCV1 vaccination coverage of 90–95% among children aged 19–35 months [12]. These outbreaks underscore the health inequities and gaps in immunization programs and primary healthcare systems [13]. This raises concerns about the coverage, timing, and factors affecting MCV2 administration [14]. Socio-demographic factors, access to healthcare, cultural beliefs, vaccine misconceptions, and mistrust in the healthcare system may contribute to low vaccination coverage [15,16]. This study aims to assess MCV2 coverage and its associated factors among children aged 19–35 months in Debre Birhan City to guide strategies for enhancing immunization and preventing future outbreaks.

## Method and materials

### Study setting, study design, and period

A community-based cross-sectional study was conducted in Debre Birhan City, North Shewa Zone, Amhara National Regional State, Ethiopia, from May 10–30, 2024. Located 130 km from Addis Ababa, the capital of Ethiopia, and 696 km from Bahir Dar, the capital of the Amhara Region. Debre Birhan is situated at an elevation of 2,840 meters above sea level, covering an area of 14.71 km$^2$. The city is divided into five sub-cities and has a population of 174,245, with 23,982 residents under the age of three [17]. Debre Birhan is served by three hospitals and eight health centers.

### Population

The source population for this study consisted of all children aged 19–35 months (birth cohort born between May 2021 to December 2022), along with their mothers or caregivers, residing in Debre Birhan City. The study population comprised randomly selected children within the same age range and their mothers or caregivers residing in this area. To be eligible for inclusion, children had to be within the specified age range (19–35 months) and have vaccination cards with recorded vaccination dates or rely on maternal recall of vaccination. Mothers or caregivers who were seriously ill and unable to participate, as well as those who had lived in the study area for less than six months, were excluded from the study.

### Sample size determination

The sample size is determined based on a single population proportion formula with a 95% level of confidence, a 5% marginal error, and P = 0.48 [18].

$n = \frac{(Z_{\frac{\alpha}{2}})^2 p(1-p)}{d^2}$, Where n= required sample size, therefore, $n = \frac{(1.96)^2 * 0.48(1-0.48)}{(0.05)2} = 384$.

By considering a 1.5 design effect and 10% non-response rate, the final sample size was 634.

### Sampling technique and procedure

A stratified sampling method, followed by systematic random sampling, was employed to select the study participants. First, two sub-cities were chosen from the available five. From these, eight kebeles (four per sub-city) were randomly selected. The sample size was allocated proportionally to each kebele based on its number of households with children aged 19–35 months.

Within each selected kebele, households with a child in this age group were chosen by systematic random sampling. The sampling interval (k) was calculated as 9 (derived from 6021 total households/ 634 required samples). In households with more than one eligible child, the youngest was selected to minimize recall bias (Fig 1).

### Variables

**Dependent variable**: The MCV2 vaccination status of the child (vaccinated/not vaccinated).

**Independent variables:** The independent variables in this study included socio-demographic factors such as child age, maternal age, maternal education, marital status, maternal occupation, paternal education, paternal occupation, monthly

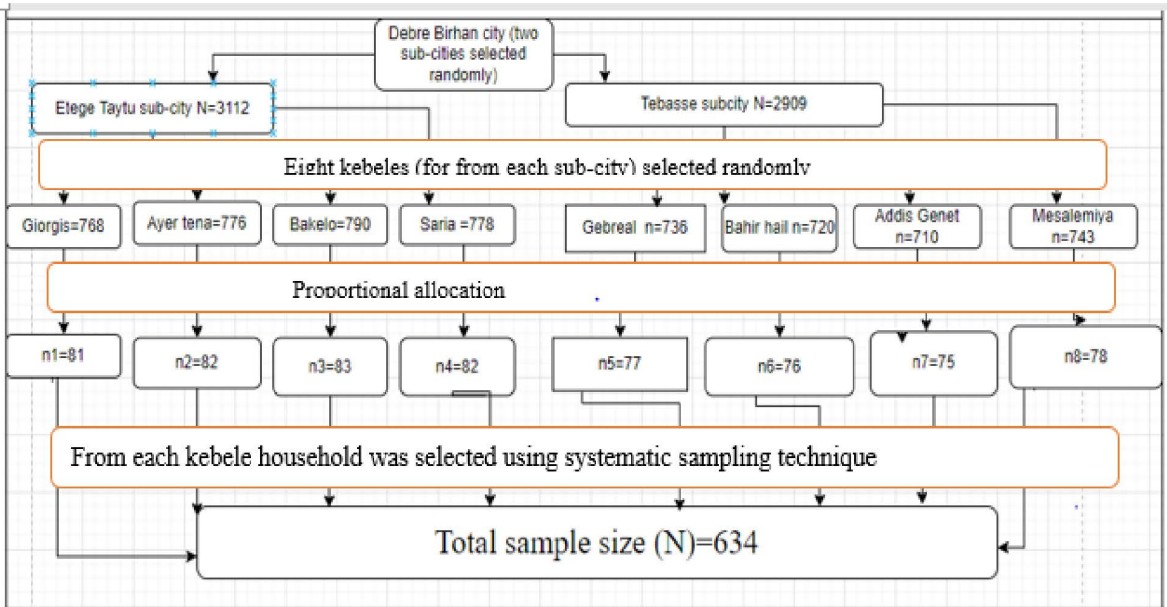

**Fig 1. Sampling procedure, 2024.**

income, family size, access to mass media, head of household, and place of residence. Maternal and child-related factors considered were antenatal care (ANC) and postnatal care (PNC) visits, tetanus toxoid (TT) vaccine uptake, birth order, place of delivery, number of children, pentavalent 3, and vitamin A uptake. Additionally, health service- and access-related factors such as travel time to the nearest facility, vaccination schedule cancellations or postponements, waiting time at the vaccination center, and vaccine availability were included. Finally, the study also considered the awareness and perception of caregivers.

## Operational definition

**MCV2 coverage**: The proportion of children aged 19–35 months who had received MCV2, regardless of when they received the MCV1, was considered to have received MCV2 [18–20].

 **Information on vaccination coverage**: was obtained in two ways; for each child aged 19–35 months, caregivers were asked to show the child's immunization card or health card used to record the child's immunizations. If vaccination was not recorded on the infant immunization card or the health card, the caregiver was asked to recall whether MCV2 vaccination had been given or not [21]. In this study, parental recall was used for 12 out of 626 children (1.9%).

 **Perception:** A set of seven perception questions were asked, and participants who scored at or above the mean were classified as having a good perception [22].

 **Awareness:** Participants who scored above or equal to the median on the seven awareness questions were considered to have a high level of awareness [20].

## Data collection method and procedures

Data were collected using Kobo Toolbox through a structured, interviewer-administered questionnaire that was adapted from prior studies [18,20,22]. The questionnaire included sections on socio-demographic characteristics, maternal and child-related factors, health service and access factors, vaccination status, and caregivers' awareness and perception of measles and its vaccine. The questionnaire initially prepared in English, and translated into the local language, Amharic,

and then back into English by different translators to ensure consistency. Prior to the main data collection, a pretest was conducted on 5% of the sample size outside the study sites. Face validity was assessed by local experts (three pedatricians), and necessary revisions were made based on their feedback. Eight BSc nurses were hired for data collection, and two MPH (Master of Public Health) supervisors were designated to oversee the process. Both data collectors and supervisors participated in a one-day training to ensure they shared a common understanding of the study's objectives, methodologies, data collection techniques, and strategies for engaging with participants. The investigators and supervisors checked the day-to-day activities of data collectors regarding the completion of questionnaires.

## Data processing and analysis

The data collected using Kobo Toolbox were subsequently exported to SPSS version 25 for coding, cleaning, and analysis. To describe the study population's characteristics, descriptive statistics such as frequencies, percentages, means, medians, standard deviations, and interquartile ranges were used. Prior to performing regression analysis, multicollinearity among independent variables was evaluated using the variance inflation factor (VIF). The Hosmer-Lemeshow goodness-of-fit test was used to assess the adequacy of the regression model (p-value of 0.83). Variables with a p-value of ≤0.2 in the bivariable analysis at a 95% confidence interval (CI) were included in the multivariable logistic regression model to account for potential confounding factors. Statistical significance was established at a p-value of <0.05. Adjusted odds ratios (AOR) and their corresponding 95% confidence intervals (CI) were calculated to determine the strength of associations between the outcome variable and the independent variables.

## Ethics approval and consent to participate

This study was reviewed and approved by the Institutional Review Board (IRB) of Debre Berhan University, Asrat Woldeyes Health Science Campus, in accordance with the ethical principles outlined in the Declaration of Helsinki and the Belmont report. The IRB approval reference number is 01/151/2024. Due to the low-risk nature of the study and contextual considerations, the IRB approved the use of verbal informed consent. Prior to participation, all caregivers were provided with clear and comprehensive information about the study's purpose, procedures, potential risks and benefits, the voluntary nature of participation, the right to withdraw at any time without penalty, and the confidentiality of their responses.

Verbal consent was documented by the data collectors on the electronic data collection platform (KoboToolbox), with a designated field to confirm that verbal consent had been obtained. In addition, the process was witnessed by a neutral third party (i.e., a community health worker) who verified and confirmed that the participant had understood the information and voluntarily agreed to participate. Permission to conduct the study was also obtained from the relevant city administration office prior to data collection.

## Results

### Socio-demographic characteristics of mother-child pairs

A total of 626 children aged 19–35 months participated in the study, with a response rate of 98.7%. The children had an mean age of 26.5 months (SD±4.7). Most respondent caregivers were married (580, 92.7%), identified as followers of the Orthodox religion (568, 90.7%), and had access to at least one functional device of TV, radio, or smartphone (496, 79.2%). Furthermore, the father was the head of the household in the majority of cases (576, 92%) (Table 1).

### Maternal and children's vaccination status

Of the 626 caregiver respondents (mothers), 565 (90.3%) had received at least two doses of the tetanus toxoid (TT) vaccine during their pregnancy with the current child. Regarding the children's basic vaccination status, 97.4% (610) had

**Table 1. Socio-demographic characteristics of mother-child pairs included in a study of second-dose measles vaccination coverage, Debre Birhan City, Ethiopia, 2024 (N=626).**

| Variables | Categories | N | % |
|---|---|---|---|
| Sex of child | Male | 302 | 48.2 |
| | Female | 324 | 51.8 |
| Residence | Urban | 306 | 48.9 |
| | Rural | 320 | 51.1 |
| Marital status of mother | Single | 13 | 2.1 |
| | Married | 580 | 92.7 |
| | Divorced | 32 | 5.1 |
| | Widowed | 1 | 0.2 |
| Mothers/caregiver Age | 19-24 | 39 | 6.2 |
| | 25-29 | 191 | 30.5 |
| | 30-35 | 250 | 39.9 |
| | >36 | 146 | 23.3 |
| Religion | Muslim | 42 | 6.7 |
| | Orthodox | 568 | 90.7 |
| | Other | 16 | 2.6 |
| Maternal educational status | Unable to read & write | 77 | 12.3 |
| | Able to read & write | 88 | 14.1 |
| | Primary | 246 | 39.3 |
| | Secondary | 145 | 23.2 |
| | College &above | 70 | 11.2 |
| Mother occupation | Housewife | 418 | 66.8 |
| | Farmer | 28 | 4.8 |
| | Business | 115 | 18.4 |
| | Gov't employee | 49 | 7.8 |
| | Others | 16 | 2.6 |
| Father educational status | Unable to read & write | 46 | 7.3 |
| | Able to read & write | 93 | 14.9 |
| | Primary | 192 | 30.7 |
| | Secondary | 182 | 29.1 |
| | College &above | 113 | 18.1 |
| Father occupation | Farmer | 236 | 37.7 |
| | Business | 220 | 35.1 |
| | Gov't employee | 119 | 19 |
| | Causal labourer | 46 | 7.3 |
| | Others | 5 | 0.8 |
| Average monthly income in Ethiopian Birr | <5000 | 184 | 29.4 |
| | 5000-10000 | 334 | 53.4 |
| | >10000 | 108 | 17.3 |
| Head of household | Mother | 50 | 8 |
| | Father | 576 | 92 |
| Family size | ≤4 | 355 | 56.7 |
| | >4 | 271 | 43.3 |
| Parity | Primipara | 149 | 23.8 |
| | Multipara | 403 | 64.4 |
| | Grandmultipara | 74 | 11.8 |

*(Continued)*

---

**Table 1.** (Continued)

| Variables | Categories | N | % |
|---|---|---|---|
| Birth order | First | 164 | 26.2 |
| | Second to fourth | 385 | 61.5 |
| | ≥Fifth birth order | 77 | 12.3 |
| The child lives with whom | Both parents | 576 | 92 |
| | Mother only | 48 | 7.7 |
| | Father only | 2 | 0.3 |
| Pregnancy status | Planned | 517 | 82.6 |
| | Unplanned | 109 | 17.4 |
| Functional radio/**television**./ Smartphone | Yes | 496 | 79.2 |
| | No | 130 | 20.8 |

received the BCG vaccine, 98.3% (615) had been given the Penta-3 vaccine, and 87.2% (546) had received the MCV1 vaccine (Table 2).

## Health service and access-related characteristics

Among the 626 child caregivers (mothers), 408 (65.1%) had attended at least four antenatal care (ANC) visits, while 23 (3.7%) had no ANC visits at all. Approximately two-thirds (67.4%) of the children were delivered at health centers. Only 222 (35.5%) of the mothers accessed postnatal care (PNC) services at least once. The median waiting time at vaccination sites was 30 minutes, with an interquartile range (IQR) of 25–35 minutes. Additionally, 89 respondents (14.2%) mentioned that their vaccination appointments had been delayed or canceled by health workers. A majority (74.1%) of the caregivers reported walking for more than 30 minutes to reach the nearest vaccination center (Table 3).

**Table 2.** Maternal and childhood vaccination-related characteristics of mother-child pairs included in a study of second-dose measles vaccination coverage, Debre Birhan City, Ethiopia, 2024 (N = 626).

| Variables | Categories | N | % |
|---|---|---|---|
| TT vaccine status | Received (two times) | 565 | 90.3 |
| | Not received | 22 | 3.5 |
| | I don't know | 39 | 6.2 |
| BCG | Vaccinated | 610 | 97.4 |
| | Not vaccinated | 16 | 2.6 |
| Penta-3 | Vaccinated | 615 | 98.2 |
| | Not vaccinated | 11 | 1.8 |
| MCV-1* | Vaccinated | 545 | 87.1 |
| | Not vaccinated | 81 | 13 |
| Vitamin A at 24 months | Not received | 234 | 37.4 |
| | At least one dose | 226 | 36.1 |
| | NA** | 166 | 26.5 |

\* All children recorded as MCV1-negative were also MCV2-negative; thus, no inconsistencies in the MCV1–MCV2 sequence were present in the dataset. NA** = not applicable meaning age 19 to less than 24 months.

**Table 3. Health service and access-related characteristics of mother-child pairs included in a study of second-dose measles vaccination coverage, Debre Birhan City, Ethiopia, 2024 (N = 626).**

| Variables | Categories | N | % |
|---|---|---|---|
| ANC contact | No contact | 23 | 3.7 |
| | 1-3 contact | 195 | 31.2 |
| | ≥ 4 contacts | 408 | 65.1 |
| PNC service utilization | Not received | 404 | 64.5 |
| | At least one | 222 | 35.5 |
| Place of delivery | Home | 16 | 2.6 |
| | Hospital | 186 | 29.7 |
| | Health center | 422 | 67.4 |
| | Private institution | 2 | 0.4 |
| Time of arrival at the nearest vaccination center on foot | <30 minute | 162 | 25.9 |
| | ≥30 minute | 464 | 74 |
| Waiting time for vaccination at the vaccination center | <30 minute | 264 | 42.2 |
| | ≥30 minute | 362 | 57.8 |
| Ever had a vaccination postponed or cancelled | Yes | 89 | 14.2 |
| | No | 537 | 85.8 |

## MCV2 coverage

The overall coverage of MCV2 among children aged 19–35 months in Debre Berhan city was 57.3% (95% CI: 53.5–61.2) (Fig 2).

## Reasons for not vaccinated

Out of 267 children who were unvaccinated for MCV2, more than two-thirds of the mothers (199, 74.5%) reported being unaware of the need for the vaccine, while 90 mothers (33.7%) cited being too busy as the reason (Fig 3).

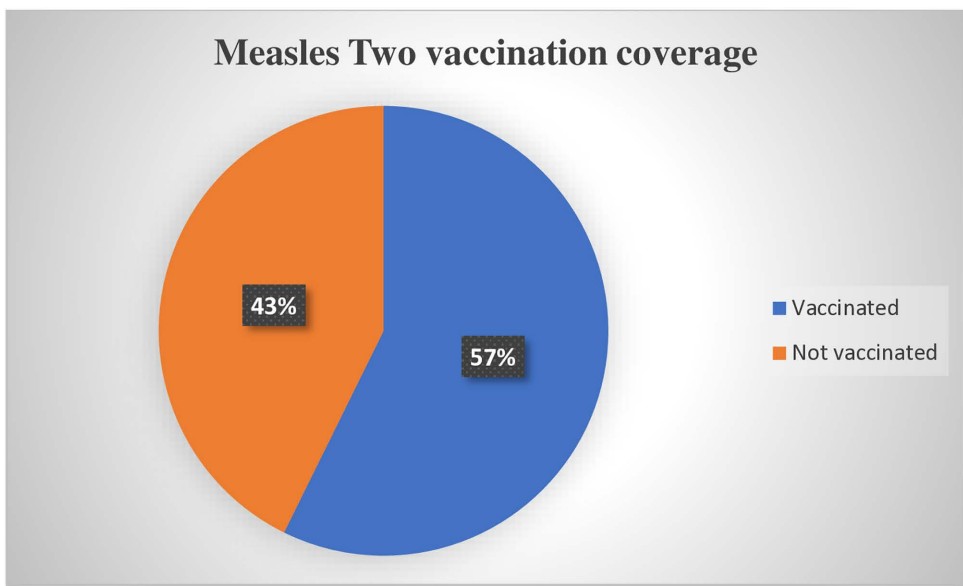

**Fig 2. Measles second-dose coverage among children aged 19–35 months in Debre Birhan City, Ethiopia, 2024.**

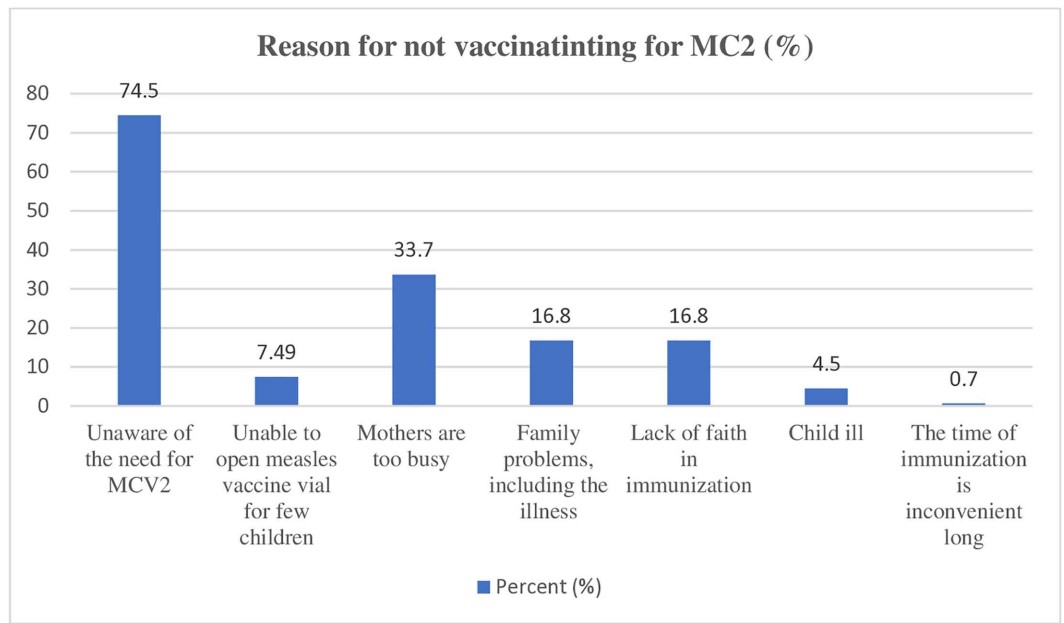

**Fig 3. Reasons for not vaccinating second dose of measles among children aged 19–35 months in Debre Birhan City, 2024.**

## Factors associated with MCV2 coverage among children

In the bivariable logistic regression analysis, twelve variables were identified with a p-value < 0.2: residence, maternal education level, father's education level, father's occupation, household headship, pregnancy status for the index child, antenatal care attendance during the current pregnancy, TT vaccination during the index pregnancy, child vaccination status, history of postponed or canceled schedules, level of awareness, and level of perception. Among these, five factors remained independently associated with having received MCV2 in the multivariable logistic regression analysis' at p<0.05. These included the father's educational status, with those able to read and write (AOR: 1.63; 95% CI: 1.17–3.69), those with primary education (AOR: 2.04; 95% CI: 1.83–7.01), and those with college education or above (AOR: 3.2; 95% CI: 1.5, 6.95); children vaccinated for any vaccine (AOR: 3.04; 95% CI: 1.33–6.98); postponed or canceled vaccination schedules (AOR: 4.9; 95% CI: 2.62–9.32); level of awareness (AOR: 3.3; 95% CI: 2.12–5.17); and level of perception (AOR: 5.1; 95% CI: 3.16–8.20)

In comparison to children whose fathers who had not received formal education, children whose father's educational status was primary, secondary, and college & above had an increased odds of receiving MCV2. The odds of receiving an MCV2 were 3 times higher for children who were vaccinated for any vaccine (AOR:3.04 (95% CI: 1.33–6.98)). In comparison to children who had a vaccination postponed or cancelled, children who had not had a previous vaccination postponed or cancelled had an increased odds of receiving MCV2 (AOR:4.9 (95% CI: 2.62–9.32)). In comparison to children whose mothers/caregiver had low awareness regarding measles vaccination, children whose mothers had high awareness had an increased odds of receiving MCV2 (AOR:3.3 (95% CI: 2.12–5.17)). In comparison to the children of mothers with a poor perception of measles vaccination, the children of mothers with a good perception of measles vaccination had an increased odds of receiving MCV2 (AOR: 5.1; 95% CI: 3.16–8.20) (Table 4).

## Discussion

Achieving measles eradication requires maintaining high levels of population immunity, which depends on ensuring the administration of two doses of the measles vaccine [23]. This study focused on assessing the coverage of the second

**Table 4. Factors associated with second-dose measles vaccination coverage among children aged 19–35 months, in Debre Birhan City, Ethiopia, 2024 (N = 626).**

| Variable | Categories | MCV-2 status | | COR (95% CI) | AOR (95%CI) | P-Value |
|---|---|---|---|---|---|---|
| | | Vaccinated | Not vaccinated | | | |
| Residence | Urban | 188 | 118 | 1.38 (1.01,1.9) | 0.65(0.38,1.12) | 0.12 |
| | Rural | 171 | 149 | 1 | 1 | |
| Maternal education | Unable to read &write | 50 | 27 | 1 | 1 | |
| | Able to read and write | 54 | 34 | 1.16 (0.61,2.2) | 1.43(0.53, 3.83) | 0.479 |
| | Primary | 101 | 145 | 2.60 (1.6,4.5) | 0.93(0.40, 2.15) | 0.861 |
| | Secondary | 45 | 100 | 4.10 (2.3,7.39) | 1.15(0.44, 2.98) | 0.77 |
| | College and above | 17 | 53 | 5.8 (2.8,11.8) | 1.04(0.33, 3.21) | 0.95 |
| Father educational status | Unable to read & write | 14 | 32 | 1 | 1 | |
| | Able to read & write | 26 | 67 | 0.89 (0.40, 1.9) | 2.1 (0.9, 2.6) | 0.053 |
| | Primary | 117 | 75 | 3.56 (1.9, 7.1) | 1.63(1.17, 3.69) | **0.024** |
| | Secondary | 116 | 66 | 4.01 (2.1, 8.1) | 2.04(1.83, 7.01) | **0.01** |
| | College and above | 86 | 27 | 7.2 (3.39, 15.6) | 3.2 (1.5, 6.95) | **0.034** |
| Father occupation | Farmer | 111 | 125 | 1 | 1 | |
| | Business | 132 | 88 | 1.7 (1.165, 2.5) | 1.32(0.75, 2.31) | 0.335 |
| | Government employed | 94 | 25 | 4.2 (2.5,7.05) | 2.07(0.75, 5.70) | 0.157 |
| | Causal labour | 22 | 29 | 0.72 (0.38,1.4) | 0.35(0.16, 1.78) | 0.053 |
| Head of household | Mother | 20 | 30 | 1 | 1 | |
| | Father | 339 | 237 | 2.15 (1.19,3.9) | 1.70(0.80, 3.62) | 0.169 |
| Pregnancy status | Planned | 304 | 213 | 1.40 (0.92,2.1) | 1.25(0.79, 2.30) | 0.278 |
| | Unplanned | 55 | 54 | 1 | 1 | |
| ANC follow up | Yes | 357 | 246 | 14.4 (3.4,62.3) | 1.9(0.35, 10.15) | 0.456 |
| | No | 2 | 21 | 1 | 1 | |
| TT Vaccination | Yes | 343 | 222 | 4.3 (2.34, 7.9) | 0.99(0.40, 2.39) | 0.958 |
| | No | 16 | 45 | 1 | 1 | |
| Child received any vaccination | Yes | 348 | 239 | 3.7 (1.8, 7.59) | 3.04(1.33, 6.98) | **0.009** |
| | No | 11 | 28 | 1 | 1 | |
| Schedules cancelled/postponed | Yes | 25 | 64 | 1 | 1 | |
| | No | 334 | 203 | 4.2 (2.57, 6.9) | 4.9 (2.62, 9.32) | **<0.001** |
| Level of awareness | low | 79 | 172 | 1 | 1 | |
| | High | 280 | 95 | 6.42 (4.5, 9.1) | 3.3 (2.1, 5.17) | **<0.001** |
| Level of perception | Poor | 47 | 147 | 1 | 1 | |
| | Good | 312 | 120 | 8.13(5.50, 12) | 5.1 (3.1, 8.20) | **<0.001** |

dose of measles vaccine and identifying factors associated with it among children aged 19–35 months in Debre Birhan City, Ethiopia.

The coverage of the second dose of the measles vaccine (MCV2) among children aged 19–35 months in Debre Birhan was 57.3% (95% CI: 53.5–61.2). This percentage is similar to those found in related studies in Indonesia, which noted a coverage rate of 54% [24] and in Kenya (56.2%) [25]. However, it is lower than the rates recorded in Gondar, Ethiopia (75.68%) [26], Zimbabwe (74%) [27], Malawi (77%) [28], China (93.9%) [29] and Japan (91%) [30].

The disparities noted, especially in comparison to China and Japan, could be attributed to differences in caregiver awareness and perceptions regarding MCV2, which may have been shaped by the timing of its integration into routine

vaccination schedules. Additionally, variations in access to healthcare services and socio-demographic factors such as maternal education, employment status, and income may also play a role in these differences.

In contrast, the coverage reported in this study exceeds the rates observed in other regions of Ethiopia, including the North Shewa Zone, Oromia Region (42.5%) [31], Jabitehnan district (48.1%) [18], Gondar City (12.36%) [32], and Kakamega district in Kenya (17.9%). This difference may be attributed to increased awareness over time, improved healthcare services, and demographic changes [19]. The lower coverage reported in North Shewa and Jabitehnan districts might be due to their predominantly rural study populations, where mothers and caregivers may have limited awareness and differing perceptions about the importance of child vaccination [18,20]. Similarly, the lower coverage observed in Gondar City in Ethiopia could be linked to the reliance on secondary data from the Ethiopian Mini Demographic and Health Survey 2019, conducted shortly after the introduction of the MCV2 vaccine [32].

The likelihood of a child receiving the MCV2 vaccination increased as the father's education level improved. Children whose fathers had a primary, secondary, or college and above education were 1.65 times (AOR: 1.63, 95% CI: 1.17–3.69), 2 times (AOR: 2.04, 95% CI: 1.83–7.01), and 3 times (AOR: 3.2, 95% CI: 1.5–6.95) more likely to be vaccinated for MCV2 compared to children of fathers who were unable to read and write. This finding aligns with survey data from six countries and a study conducted in Turkey, both of which demonstrated a positive correlation between fathers' formal education and the likelihood of their children being vaccinated for measles [33,34]. This association may be explained by the fact that fathers are the heads of most households in the study area (92%). Notably, this study found no significant association between mothers' education and MCV2 vaccination, whereas other studies have demonstrated a strong link between higher maternal education levels and improved MCV2 immunization rates [35,36].

Children who had previously received any vaccine were three times more likely to receive MCV2 compared to their counterparts (AOR: 3.04; 95% CI: 1.33–6.98). This finding aligns with studies conducted in Ethiopia, including Gondar, and Kenya [22,37]. A possible explanation is that children who are already vaccinated may have better access to healthcare, and their parents may be more knowledgeable about the vaccination schedule and the importance of the MCV2 vaccine [22,37,38]. Regular interactions with healthcare professionals during vaccination services likely encourage caregivers to complete their children's vaccinations. Additionally, frequent visits to health facilities for vaccinations may expose caregivers to positive examples of others vaccinating their children, further reinforcing the practice [39,40].

Furthermore, mothers or caregivers who did not have any vaccination schedules canceled or postponed were nearly five times more likely to vaccinate their children for MCV2 compared to those who experienced schedule disruptions (AOR: 4.9; 95% CI: 2.62–9.32). This result is consistent with studies from the West Gojjam zone in Ethiopia and Ballabgarh in India [41,42].

Mothers or caregivers with a high level of awareness about measles vaccination were three times more likely to vaccinate their children with MCV2 than those with low awareness (AOR: 3.3 (95% CI: 2.12–5.17)). Similarly, the likelihood of vaccinating their children with MCV2 was five times higher (AOR:5.1, 95% CI:3.16–8.20) among children of mothers or caregivers who had a good perception of measles vaccination compared to those with a poor perception. This finding aligns with studies conducted in Ethiopia; Gondar, Jabitehnan district, Ormia region North shewa and Kenya [18,26,31,43]. The increased likelihood may be attributed to mothers or caregivers who have a better understanding of vaccine-preventable diseases, the number of recommended measles doses, and the timing of each vaccination, leading to a greater commitment to vaccinating their children [44]. Furthermore, if the mother or caregiver perceives the severity of measles and the benefits of the measles vaccine positively, she is more likely to intend to vaccinate her child [35].

Beyond the above factors, to improve the coverage of the second dose of measles vaccination among children aged 19–35 months, it is essential to implement context-specific, evidence-based strategies. One key approach is to enhance community awareness through regular outreach activities and caregiver education programs and locally tailored media campaigns. Studies in Ethiopia and other low-resource settings have shown that community-based health education significantly increases vaccine uptake by improving caregiver knowledge and attitudes toward immunization [45] Additionally,

introducing or strengthening reminder systems, either electronic (e.g., mobile phone text messages) or follow-up visits by community health workers further reduce dropout rates by addressing forgetfulness or access barriers [46].

## Limitations of the study

A key limitation of this study is that vaccination status for some children was determined based on caregiver recall due to the absence of vaccination cards, which may introduce recall bias. Furthermore, verification of vaccination records at the facility level was not conducted, and such verification could have enhanced the accuracy and reliability of the reported vaccination status.

## Conclusion and recommendation

The coverage of MCV2 in the study area is significantly lower than the World Health Organization and the national target. Key factors influencing coverage include the father's educational status, prior vaccination, uninterrupted vaccination schedules, and caregiver awareness and perception. These findings suggest that health workers should focus on awareness creation about MCV2 vaccine schedules.. Also, efforts should be made to reduce possible obstacles such as cancellation/postponement of the vaccination schedule.

## Supporting information

**S1 Data set. Dataset for cross-checking the analysis.** Excel file containing the underlying dataset used for the analyses in this study; provided in response to reviewer request for verification purposes.
(CSV)

## Author contributions

**Conceptualization:** Tebabere Moltot, Eyosias Yeshialem.

**Data curation:** Tebabere Moltot, Zenebe Abebe Gebreegziabher, Agmasie Damtew Walle, Mistire Nigussie, Addisalem Workie Demsash, Eyosias Yeshialem.

**Formal analysis:** Tebabere Moltot, Zenebe Abebe Gebreegziabher, Agmasie Damtew Walle, Micheal Amera Tizazu, Addisalem Workie Demsash, Eyosias Yeshialem.

**Funding acquisition:** Mistire Nigussie.

**Investigation:** Tebabere Moltot, Micheal Amera Tizazu.

**Methodology:** Tebabere Moltot, Zenebe Abebe Gebreegziabher, Agmasie Damtew Walle, Addisalem Workie Demsash, Eyosias Yeshialem.

**Resources:** Tebabere Moltot, Mistire Nigussie, Eyosias Yeshialem.

**Software:** Tebabere Moltot, Zenebe Abebe Gebreegziabher, Agmasie Damtew Walle, Micheal Amera Tizazu, Addisalem Workie Demsash, Eyosias Yeshialem.

**Supervision:** Tebabere Moltot, Zenebe Abebe Gebreegziabher, Agmasie Damtew Walle, Micheal Amera Tizazu, Addisalem Workie Demsash.

**Validation:** Tebabere Moltot, Zenebe Abebe Gebreegziabher, Agmasie Damtew Walle, Mistire Nigussie, Micheal Amera Tizazu, Addisalem Workie Demsash, Eyosias Yeshialem.

**Visualization:** Zenebe Abebe Gebreegziabher, Mistire Nigussie, Micheal Amera Tizazu, Addisalem Workie Demsash, Eyosias Yeshialem.

**Writing – original draft:** Tebabere Moltot, Micheal Amera Tizazu.

**Writing – review & editing:** Tebabere Moltot, Zenebe Abebe Gebreegziabher, Agmasie Damtew Walle, Mistire Nigussie, Micheal Amera Tizazu, Addisalem Workie Demsash, Eyosias Yeshialem.

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
