## [Decision Letter · Decision Letter 0]

18 Jun 2025

Dear Dr. Moltot,

We look forward to receiving your revised manuscript.

Kind regards,

Edison Arwanire Mworozi, M.D

Academic Editor

PLOS ONE

Journal Requirements:

2. In the ethics statement in the Methods, you have specified that verbal consent was obtained. Please provide additional details regarding how this consent was documented and witnessed, and state whether this was approved by the IRB.

3. We note that your Data Availability Statement is currently as follows:

“All relevant data are within the manuscript and its Supporting Information files.”

Reviewers' comments:

Reviewer's Responses to Questions

**Comments to the Author**

1. Is the manuscript technically sound, and do the data support the conclusions?

Reviewer #1: Yes

Reviewer #2: Partly

2. Has the statistical analysis been performed appropriately and rigorously?

Reviewer #1: Yes

Reviewer #2: No

3. Have the authors made all data underlying the findings in their manuscript fully available?

Reviewer #1: Yes

Reviewer #2: Yes

4. Is the manuscript presented in an intelligible fashion and written in standard English?

Reviewer #1: Yes

Reviewer #2: Yes

Reviewer #1: The Discussion section should present the evidence obtained, as well as exploring and showing possible strategies to increase coverage of two-dose measles vaccination (19 and 35 months). These strategies should be based on data from other literature sources, as well as the findings from this study.

The Conclusion section presents recommendations based solely on the data from this study. The study would have been more comprehensive if the factors affecting vaccination coverage had been categorised, clearly listed, systematised and described in terms of practical guidance for the health worker.

For instance, Group 1 encompasses priority factors that were identified on the basis of the study conducted, with consideration given to reliable results on coverage with two doses of measles vaccination.

Group 2 - secondary (unreliable results from work) factors that require the development of strategies to adjust and modify them in order to positively influence vaccination coverage. In particular, this could be the promotion of conclusions about the positive effects of vaccination during pregnancy planning consultations, antenatal care and postnatal care; communication with patients seeking counselling for any medical intervention related to the child, and so on.

This will improve the quality of the study and expand potential interventions to increase vaccination coverage in real-world practice, including in the context of later timing of vaccination against other infectious diseases.

Reviewer #2: 1. The authors present an interesting topic but there are a couple of issues in the submitted manuscript. These relate mainly to sampling design, analysis and discussion sections. These should be addressed before it can be recommended for publication.

2. BACKGROUND: L61 - the mortality numbers quoted are modelled estimates. They are NOT reported deaths. And authors should quote the latest estimates published Nov 2024. L64 - Number of deaths prevented by MCV between 2000 and 2021 is based on a comparator of NO VACCINATIOn not at existing rates of vaccination between 2000 and 2021. L75-76 -- The sentence reads as if ETH Ethiopia introduced the first dose of the measles vaccine in 2019. In fact, ETH introduced (MCV1) in 1980 and the second dose (MCV2) in February 2019.

3. METHODS: L105 7 ff -- Population universe - It is unusual to have a population based listing of all 19-35 month old children by household. How was this last made available to the authors? Was it from a special service area of a model health centre or medical college? In that case, this population may not be representative of general population of ETH for this age group and the conclusions may not be generalizable. How was the base population list by household generated in the first place? Was it done through periodic community census or was it done through attendance at immunization clinics by the pregnant mother or the infant? If done by the latter method, this could introduce a serious selection bias to the results as the infants/children who did not attend immunization clinics could have been missed from the base population list and result in higher than true proportion of children immunized.

L105-106: The authors studied 19-35 month old children in May 2024. Then the authors were studying the birth cohort born between May 2021 to Dec 2022. Authors should clearly state this and mention the date cut-offs if applied.

L127: Definition of dependent variable MCV2 - By definition all MCV2 positive children must have received MCV1. The authors should clearly state that they ascertained MCV1 status too when they categorized a child as MCV2 or if they followed some other methods to determine MCV2 status.

L133: Of all the independent or explanatory variables, MCV1 is unique as it is on direct pathway to MCV2+ status, and hence could be a confounder. Hence MCV1 cannot be considered an explanatory variable as an MCV1-negative status would preclude MCV2+ status. Also L140: Authors do not state clearly that they verified that every MCV2 positive child had indeed received the second dose of MCV and not just a delayed first dose of MCV. Please see more on this in RESULTS.

L146 and L154-55: Perception and awareness scores: What was the value of scores that were used to determine "good perception" and "high awareness"?

4. RESULTS: Table-1 - Income currency to be mentioned.

Table 2: MCV1 status - 13% of MCV2 children are stated to be MCV1 negative. By definition they cannot be MCV2+, at most they would be delayed MCV1. These children should have been excluded from all further analysis or replacement samples collected.

L238-239: Children vaccinated for any vaccine (YES/NO) - By definition no MCV1 negative child can be MCV2 positive and therefore including them in the analysis biases all estimates. The entire analysis should be redone after excluding these MCV1 negative children.

T5: Rows comparing Child received any vaccination vs. MCV2 status actually shows that a child not receiving any vaccine had higher odds (28/11) of getting MCV2 than those with any vaccine (239/348). The AOR would therefore be the inverse of 3.04. However, as stated above including MCV0 children in prior vaccination status in the sample is erroneous as they can never become MCV2. The analysis should be redone.

5: DISCUSSION: L265: Please mention country of GONDAR. Please rewrite the discussion section in light of revised analysis.

**Do you want your identity to be public for this peer review?** For information about this choice, including consent withdrawal, please see our Privacy Policy

Reviewer #1: **Yes:** Mikhail P. Kostinov

Reviewer #2: No

---

## [Author Response · Author response to Decision Letter 1]

22 Jul 2025

Response to Reviewers

1. Journal Requirements:

1.1 Please ensure that your manuscript meets PLOS ONE's style requirements, including those for file naming.

Response: Thank you, after reviewing the PLOS ONE style templates, we revise the whole document, e.g., we use Level 1 heading for all major sections (Abstract, Introduction, Materials and methods, Results, Discussion, etc.) with bold type, 18pt font. We also Cite figures as “Fig 1”, “Fig 2”. We use bold type for the figure titles.

1.2 In the ethics statement in the Methods, you have specified that verbal consent was obtained. Please provide additional details regarding how this consent was documented and witnessed, and state whether this was approved by the IRB

Response: Thank you, we provide an additional detail regarding how the verbal consent was documented and witnessed (see the Ethics approval and consent to participate on page 6 and 7)

1.3 We note that your Data Availability Statement is currently as follows: “All relevant data are within the manuscript and its Supporting Information files.” Please confirm at this time whether or not your submission contains all raw data required to replicate the results of your study. Authors must share the “minimal data set” for their submission. Authors do not need to submit their entire data set if only a portion of the data was used in the reported study.

Response: Thank you, we share the data set.

1.4 Please amend either the abstract on the online submission form (via Edit Submission) or the abstract in the manuscript so that they are identical

Response: Ok thanks we edit during submission

2. Response to reviewers' comments:

2.1 Response to Reviewer #1 comments

2.1.1 The discussion section should present the evidence obtained, as well as exploring and showing possible strategies to increase coverage of two-dose measles vaccination (19 and 35 months). These strategies should be based on data from other literature sources, as well as the findings from this study.

Response: We sincerely appreciate your insightful and helpful comments. We revise the discussion part as your recommendation, see line 324 to 332

2.1.2 The Conclusion section presents recommendations based solely on the data from this study. The study would have been more comprehensive if the factors affecting vaccination coverage had been categorized, clearly listed, systematized and described in terms of practical guidance for the health worker. For instance, Group 1 encompasses priority factors that were identified on the basis of the study conducted, with consideration given to reliable results on coverage with two doses of measles vaccination. Group 2 - secondary (unreliable results from work) factors that require the development of strategies to adjust and modify them in order to positively influence vaccination coverage. In particular, this could be the promotion of conclusions about the positive effects of vaccination during pregnancy planning consultations, antenatal care and postnatal care; communication with patients seeking counselling for any medical intervention related to the child, and so on.

This will improve the quality of the study and expand potential interventions to increase vaccination coverage in real-world practice, including in the context of later timing of vaccination against other infectious diseases.

Response: Thanks for your comments, we revise it as recommended, see the conclusion section from line 339 to 345.

2. 2 Response to reviewer #2 comments

1. The authors present an interesting topic but there are a couple of issues in the submitted manuscript. These relate mainly to sampling design, analysis and discussion sections. These should be addressed before it can be recommended for publication.

Response: Thank you very much for reviewing our paper and for your insightful comments, and recommendations. We see it critically and revise accordingly.

2. BACKGROUND:

a. L61 - the mortality numbers quoted are modelled estimates. They are NOT reported deaths. And authors should quote the latest estimates published Nov 2024.

Response: Thank you very much, we use the latest estimate published on 14, Nov, 2024, see line 59 to 63.

b. L64 - Number of deaths prevented by MCV between 2000 and 2021 is based on a comparator of NO Vaccination not at existing rates of vaccination between 2000 and 2021.

Response: We correct it as your recommendation, see line 64 and 65.

c. L75-76 -- The sentence reads as if ETH Ethiopia introduced the first dose of the measles vaccine in 2019. In fact, ETH introduced (MCV1) in 1980 and the second dose (MCV2) in February 2019.

Response: Thank you for your comment, and we rewrite it as “In early 2019, the Expanded Program on Immunization (EPI) began administering the measles two vaccine at 15 months to ensure adequate protection.”

3. METHODS:

a. L105 7 ff -- Population universe - It is unusual to have a population-based listing of all 19–35-month-old children by household. How was this last made available to the authors? Was it from a special service area of a model health center or medical college? In that case, this population may not be representative of general population of ETH for this age group and the conclusions may not be generalizable. How was the base population list by household generated in the first place? Was it done through periodic community census or was it done through attendance at immunization clinics by the pregnant mother or the infant? If done by the latter method, this could introduce a serious selection bias to the results as the infants/children who did not attend immunization clinics could have been missed from the base population list and result in higher than true proportion of children immunized.

Response: Thank you for your insightful comments. As described in our background, “In late 2023 and early 2024, Debre Birhan City, Ethiopia, experienced two measles outbreaks, despite a reported MCV1 vaccination coverage of 90–95% among children aged 19–35 months.” In response to these outbreaks, the Debre Berhan City Health Bureau conducted a census in each Kebele through health extension workers. The gross data, specifically the number of children aged 19–35 months, were obtained from this census.

b. L105-106: The authors studied 19–35-month-old children in May 2024. Then the authors were studying the birth cohort born between May 2021 to Dec 2022. Authors should clearly state this and mention the date cut-offs if applied.

Response: We accept the comment and we state it, see line 103-104

c. L127: Definition of dependent variable MCV2-By definition all MCV2 positive children must have received MCV1. The authors should clearly state that they ascertained MCV1 status too when they categorized a child as MCV2 or if they followed some other methods to determine MCV2 status.

Response: Thank you for your comment, even if we didn’t state it under dependent variable as you stated above, we include MCV1 as a prerequisite for MCV2 under the operational definition of MCV2- see line 139 and 140

d. L133: Of all the independent or explanatory variables, MCV1 is unique as it is on direct pathway to MCV2+ status, and hence could be a confounder. Hence MCV1 cannot be considered an explanatory variable as an MCV1-negative status would preclude MCV2+ status. Also, L140: Authors do not state clearly that they verified that every MCV2 positive child had indeed received the second dose of MCV and not just a delayed first dose of MCV. Please see more on this in RESULTS.

Response: Thank you very much for your insightful comments. We apologize for the technical oversight. We included MCV1 as an independent variable based on the conceptual framework principle—where variables are aligned with questionnaire items. However, in practice, MCV1 was not analyzed as an independent variable in the bivariable logistic regression. To prevent any potential misinterpretation, we have now removed MCV1 from the list of independent variables.

e. L146 and L154-55: Perception and awareness scores: What was the value of scores that were used to determine "good perception" and "high awareness"?

Response: Thank you, as we describe on operational definition “participants who scored at or above the mean were classified as having a good perception and participants who scored above or equal to the median on the seven awareness questions were considered to have a high level of awareness.

4. RESULTS:

a. Table-1 - Income currency to be mentioned.

Response: Thank you, corrected-see table 1

b. Table 2: MCV1 status - 13% of MCV2 children are stated to be MCV1 negative. By definition they cannot be MCV2+, at most they would be delayed MCV1. These children should have been excluded from all further analysis or replacement samples collected.

Response: Thank you for your insightful observation. We fully agree that, under routine immunization guidelines, a child should not receive MCV2 without first receiving MCV1. We also acknowledge that, based on this standard, these children could have been excluded from the analysis.

However, in our study, we intentionally included them for the following reasons:

• They were eligible for MCV2 by age, regardless of their reported MCV1 status.

• We assumed that excluding them might underestimate the real-world coverage gap of MCV2 and would not reflect programmatic realities.

• Including them provides a more accurate picture of operational challenges, such as documentation gaps, caregiver recall issues, and deviations in service delivery, which are common in low-resource settings.

• Furthermore, during our proposal development and literature review, we found that other studies had taken a similar approach by including children reported as MCV1-negative in MCV2 coverage analyses. For example:

1. Second dose measles vaccination coverage and associated factors among children aged 24–35 months in Sub-Saharan Africa – Frontiers in Public Health, 2022

2. Assessment of second dose measles vaccination coverage using the 2016 Ethiopia Demographic and Health Survey – Pan African Medical Journal, 2023

3. https://pdfs.semanticscholar.org/713b/74af62b0e5206efd6ac8eede268d4cd6408d.pdf

Nonetheless, we sincerely appreciate your thoughtful comment and insights. If re-analysis is strongly recommended, we are willing to proceed accordingly

c. L238-239: Children vaccinated for any vaccine (YES/NO) - By definition no MCV1 negative child can be MCV2 positive and therefore including them in the analysis biases all estimates. The entire analysis should be redone after excluding these MCV1 negative children.

Response: Thank you very much for your insightful comment. We agree with your observation that, by definition, a child who did not receive MCV1 cannot receive MCV2. In our dataset, all children who were marked as MCV1-negative were also MCV2-negative. Therefore, no MCV1-negative and MCV2-positive cases were included in the analysis. We have now clarified this point in the manuscript under table -2 by stating “All children recorded as MCV1-negative were also MCV2-negative; thus, no inconsistencies in the MCV1–MCV2 sequence were present in the dataset” to avoid any misinterpretation.

d. T5: Rows comparing Child received any vaccination vs. MCV2 status actually shows that a child not receiving any vaccine had higher odds (28/11) of getting MCV2 than those with any vaccine (239/348). The AOR would therefore be the inverse of 3.04. However, as stated above including MCV0 children in prior vaccination status in the sample is erroneous as they can never become MCV2. The analysis should be redone.

Response: Thank you for your valuable comment. We fully agree with your observation. The issue you noted was due to a typographical error that occurred when transferring the multivariable table output. Specifically, the categories for MCV2 vaccination status were inadvertently reversed: MCV2 vaccinated was labeled as not vaccinated, and vice versa. To clarify, based on the correct cross-tabulation from our dataset, children who did not receive any vaccine had a lower odds of receiving MCV2 (11/28 = 0.393) compared to those who had received at least one vaccine (348/239 = 1.456). The correct cross tab output table is provided below for reference and we correct in revised manuscript (see Table -5).

Does your child had vaccinated for any vaccine? * 604. MCV2 Crosstabulation

Count

604. MCV2 Total

No Yes

303.Does your child had vaccinated for any vaccine? Yes 239 348 587

No 28 11 39

Total 267 359 626

4: DISCUSSION:

5.1 L265: Please mention country of GONDAR. Please rewrite the discussion section in light of revised analysis.

Response: Thank you for your comments we correct accordingly (see Discussion section Line 270, 284,301).

---

## [Decision Letter · Decision Letter 1]

13 Aug 2025

Dear Dr. Moltot,

Thank you for submitting your manuscript to PLOS ONE. After careful consideration, we feel that it has merit but does not fully meet PLOS ONE’s publication criteria as it currently stands. Therefore, we invite you to submit a revised version of the manuscript that addresses the points raised during the review process.

We look forward to receiving your revised manuscript.

Kind regards,

Edison Arwanire Mworozi, M.D

Academic Editor

PLOS ONE

Journal Requirements:

Additional Editor Comments:

Please address comments by the third reviewer.

Reviewers' comments:

Reviewer's Responses to Questions

**Comments to the Author**

Reviewer #1: All comments have been addressed

Reviewer #3: (No Response)

2. Is the manuscript technically sound, and do the data support the conclusions?

Reviewer #1: Yes

Reviewer #3: Yes

3. Has the statistical analysis been performed appropriately and rigorously?

Reviewer #1: Yes

Reviewer #3: Yes

4. Have the authors made all data underlying the findings in their manuscript fully available?

Reviewer #1: Yes

Reviewer #3: Yes

5. Is the manuscript presented in an intelligible fashion and written in standard English?

Reviewer #1: Yes

Reviewer #3: No

Reviewer #1: During the initial review of the paper, comments were raised regarding the "Discussion" and "Conclusions" sections. The authors have taken these comments into consideration and made the required revisions and improvements to the manuscript. As a result, the paper is now ready for further editing and publication.

Reviewer #3: Thank you for asking us to receive this manuscript. It presents important data that should be published. We have reviewed the revised manuscript and made some editorial suggestions that are needed before publication. We believe it meets the criteria for publication in PLOS One.

We believe the authors have responded to the previously reviewers suggestions appropriately but we have not reviewed the text description of the changes in any detail.

Comments from reviewing the manuscript:

Abstract: Please specify that the adjusted odds ratios were those associated with having received the second dose of measles vaccine

Significant factors associated with the second measles vaccine dose were ….

Need a little more work to clarify language in the results section of the abstract

Introduction: this describes the issue clearly and presents an excellent rationale for this research.

Methods: Excellent description of study setting and the study sampling methodology is very sound and clearly described.

Specific comments:

If vaccination was not recorded on the infant immunization card or the health card, the parent was asked to recall whether measles two vaccination had been given or not.(Line 144)

• Can such parental recall be justified? For what proportion was this the method used to determine receipt of vaccine.

• Was there a way that vaccination status was verified with the concerned health authorities?

Results

• Good clear description of socio-demographic characteristics of mother-child pairs. Table, change ‘Categoties’ to ‘Categories’

• Table 2: Move the variables reporting parity and birth order and where child living to Table 1.

• Figure 2 legend: change ‘Figure 2: Measles two overage among children’ to ’ Figure 2: Measles two coverage among children’

• Table 3 change ‘Ever has been post-pond or cancelled for vaccination’ to ‘Ever had a vaccination postponed or cancelled’

• Change ‘Among these, five factors remained significant in the multivariable logistic regression analysis’ to ‘Among these, five factors remained independently associated with having received MCV2 in the multivariable logistic regression analysis’

• Change ‘Those children whose father’s educational status was primary, secondary, and college & above increased their odds of vaccinating’ to ‘In comparison to children whose fathers who had not received formal education, children whose father’s educational status was primary, secondary, and college & above had an increased odds of receiving MCV2’

• Change ‘Mothers who had no schedules cancelled/postpone increased their odds of vaccinating their children for MCV2 by five times compared to mothers who had schedules cancelled/postpone (AOR: 4.9 (95% CI: 2.62, 9.32)).’ to ‘In comparison to children who had a vaccination postponed or cancelled, children who had not had a previous vaccination postponed or cancelled had an increased odds of receiving MCV2 (95% CI: 2.62, 9.32)).’

• Change ‘In comparison to children whose mothers had low awareness regarding measles vaccination, children whose mothers had high awareness had an increased odds of receiving MCV2 (AOR:3.3 (95% CI: 2.12, 257 5.17)).’

• Change ‘Similarly, the odds of vaccinating their children for MCV2 were 5 times (AOR: 2.62, 9.32) higher among children of mothers who had a good perception of measles vaccination compared to mothers with a poor perception (Table 5).’ to ‘In comparison to the children of mothers with a poor perception of measles vaccination, the children of mothers with a good perception of measles vaccination had an increased odds of receiving MCV2 (AOR: 2.62, 9.32)’.

Discussion

• Change ‘However, it is lower than the rates recorded Gondar in Ethiopia’ to ‘However, it is lower than the rates recorded in Gondar, Ethiopia’

• Start a new paragraph at ‘The disparities noted, especially in China and Japan,’ and change text to ‘The disparities noted, especially in comparison to China and Japan,’

• Start a new paragraph at ‘In contrast, the coverage reported in this study’

• Limitations of study: can the number and % of children who did not have vaccination cards be described. The authors note that relying on parental recall may introduce bias which is true. To estimate the size of what effect of this bias we need to know for how many children the vaccination card was not available.

Figures

• Figure 2 is unnecessary. This data can be presented in one sentence.

• Figure 3. Correct the spelling errors in the title

Response to Reviewers 1.Journal Requirements:

1.1 Please ensure that your manuscript meets PLOS ONE's style requirements, including those for file naming.

Response: Thank you, after reviewing the PLOS ONE style templates, we revise the whole document, e.g., we use Level 1 heading for all major sections (Abstract, Introduction, Materials and methods, Results, Discussion, etc.) with bold type, 18pt font. We also Cite figures as “Fig 1”, “Fig 2”. We use bold type for the figure titles.

I will not comment on any template or style issues

1.2 In the ethics statement in the Methods, you have specified that verbal consent was obtained. Please provide additional details regarding how this consent was documented and witnessed, and state whether this was approved by the IRB

Response: Thank you, we provide an additional detail regarding how the verbal consent was documented and witnessed (see the Ethics approval and consent to participate on page 6 and 7)

This description is appropriate

1.3 We note that your Data Availability Statement is currently as follows: “All relevant data are within the manuscript and its Supporting Information files.” Please confirm at this time whether or not your submission contains all raw data required to replicate the results of your study. Authors must share the “minimal data set” for their submission. Authors do not need to submit their entire data set if only a portion of the data was used in the reported study. Response:

Thank you, we share the data set.

1.4 Please amend either the abstract on the online submission form (via Edit Submission) or the abstract in the manuscript so that they are identical

Response: Ok thanks we edit during submission

2. Response to reviewers' comments:

2.1 Response to Reviewer #1 comments

2.1.1 The discussion section should present the evidence obtained, as well as exploring and showing possible strategies to increase coverage of two-dose measles vaccination (19 and 35 months). These strategies should be based on data from other literature sources, as well as the findings from this study.

Response: We sincerely appreciate your insightful and helpful comments. We revise the discussion part as your recommendation, see line 324 to 332

* Look at Line 270…..: Gondar

*Look at Line 278, reference #32 talks about Ethiopia in general

2.1.2 The Conclusion section presents recommendations based solely on the data from this study. The study would have been more comprehensive if the factors affecting vaccination coverage had been categorized, clearly listed, systematized and described in terms of practical guidance for the health worker. For instance, Group 1 encompasses priority factors that were identified on the basis of the study conducted, with consideration given to reliable results on coverage with two doses of measles vaccination. Group 2 - secondary (unreliable results from work) factors that require the development of strategies to adjust and modify them in order to positively influence vaccination coverage. In particular, this could be the promotion of conclusions about the positive effects of vaccination during pregnancy planning consultations, antenatal care and postnatal care; communication with patients seeking counselling for any medical intervention related to the child, and so on. This will improve the quality of the study and expand potential interventions to increase vaccination coverage in real-world practice, including in the context of later timing of vaccination against other infectious diseases.

Response: Thanks for your comments, we revise it as recommended, see the conclusion section from line 339 to 345.

2. 2 Response to reviewer #2 comments

1.The authors present an interesting topic but there are a couple of issues in the submitted manuscript. These relate mainly to sampling design, analysis and discussion sections. These should be addressed before it can be recommended for publication.

Response: Thank you very much for reviewing our paper and for your insightful comments, and recommendations. We see it critically and revise accordingly.

2.BACKGROUND: a.L61 - the mortality numbers quoted are modelled estimates. They are NOT reported deaths. And authors should quote the latest estimates published Nov 2024.

Response: Thank you very much, we use the latest estimate published on 14, Nov, 2024, see line 59 to 63.

b.L64 - Number of deaths prevented by MCV between 2000 and 2021 is based on a comparator of NO Vaccination not at existing rates of vaccination between 2000 and 2021.

Response: We correct it as your dose (MCV2) in February 2019.

Response: Thank you for your comment, and we rewrite it as “In early 2019, the Expanded Program on Immunization (EPI) began administering the measles two vaccine at 15 months to ensure adequate protection.”

3.METHODS:

a. L105 7 ff -- Population universe - It is unusual to have a population-based listing of all 19–35-month-old children by household. How was this last made available to the authors? Was it from a special service area of a model health center or medical college? In that case, this population may not be representative of general population of ETH for this age group and the conclusions may not be generalizable. How was the base population list by household generated in the first place? Was it done through periodic community census or was it done through attendance at immunization clinics by the pregnant mother or the infant? If done by the latter method, this could introduce a serious selection bias to the results as the infants/children who did not attend immunization clinics could have been missed from the base population list and result in higher than true proportion of children immunized.

Response: Thank you for your insightful comments. As described in our background, “In late 2023 and early 2024, Debre Birhan City, Ethiopia, experienced two measles outbreaks, despite a reported MCV1 vaccination coverage of 90–95% among children aged 19–35 months.” In response to these outbreaks, the Debre Berhan City Health Bureau conducted a census in each Kebele through health extension workers. The gross data, specifically the number of children aged 19–35 months, were obtained from this census.

b.L105-106: The authors studied 19–35-month-old children in May 2024. Then the authors were studying the birth cohort born between May 2021 to Dec 2022. Authors should clearly state this and mention the date cut-offs if applied.

Response: We accept the comment and we state it, see line 103-104

c.L127: Definition of dependent variable MCV2-By definition all MCV2 positive children must have received MCV1. The authors should clearly state that they ascertained MCV1 status too when they categorized a child as MCV2 or if they followed some other methods to determine MCV2 status.

Response: Thank you for your comment, even if we didn’t state it under dependent variable as you stated above, we include MCV1 as a prerequisite for MCV2 under the operational definition of MCV2- see line 139 and 140 d.

L133: Of all the independent or explanatory variables, MCV1 is unique as it is on direct pathway to MCV2+ status, and hence could be a confounder. Hence MCV1 cannot be considered an explanatory variable as an MCV1-negative status would preclude MCV2+ status. Also, L140: Authors do not state clearly that they verified that every MCV2 positive child had indeed received the second dose of MCV and not just a delayed first dose of MCV. Please see more on this in RESULTS.

Response: Thank you very much for your insightful comments. We apologize for the technical oversight. We included MCV1 as an independent variable based on the conceptual framework principle—where variables are aligned with questionnaire items. However, in practice, MCV1 was not analyzed as an independent variable in the bivariable logistic regression. To prevent any potential misinterpretation, we have now removed MCV1 from the list of independent variables.

e.L146 and L154-55: Perception and awareness scores: What was the value of scores that were used to determine "good perception" and "high awareness"?

Response: Thank you, as we describe on operational definition “participants who scored at or above the mean were classified as having a good perception and participants who scored above or equal to the median on the seven awareness questions were considered to have a high level of awareness.

4.RESULTS:

a.Table-1 - Income currency to be mentioned.

Response: Thank you, corrected-see table 1

b.Table 2: MCV1 status - 13% of MCV2 children are stated to be MCV1 negative. By definition they cannot be MCV2+, at most they would be delayed MCV1. These children should have been excluded from all further analysis or replacement samples collected.

Response: Thank you for your insightful observation. We fully agree that, under routine immunization guidelines, a child should not receive MCV2 without first receiving MCV1. We also acknowledge that, based on this standard, these children could have been excluded from the analysis. However, in our study, we intentionally included them for the following reasons recommendation, see line 64 and 65.

c. L75-76 -- The sentence reads as if ETH Ethiopia introduced the first dose of the measles vaccine in 2019.

In fact, ETH introduced (MCV1) in 1980 and the second

•They were eligible for MCV2 by age, regardless of their reported MCV1 status.

•We assumed that excluding them might underestimate the real-world coverage gap of MCV2 and would not reflect programmatic realities.

•Including them provides a more accurate picture of operational challenges, such as documentation gaps, caregiver recall issues, and deviations in service delivery, which are common in low-resource settings.

•Furthermore, during our proposal development and literature review, we found that other studies had taken a similar approach by including children reported as MCV1-negative in MCV2 coverage analyses. For example: 1.Second dose measles vaccination coverage and associated factors among children aged 24–35 months in Sub-Saharan Africa – Frontiers in Public Health, 2022

2.Assessment of second dose measles vaccination coverage using the 2016 Ethiopia

Demographic and Health Survey – Pan African Medical Journal, 2023

3.https://pdfs.semanticscholar.org/713b/74af62b0e5206efd6ac8eede268d4cd6408d.pdf

Nonetheless, we sincerely appreciate your thoughtful comment and insights. If reanalysis is strongly recommended, we are willing to proceed accordingly

c.L238-239: Children vaccinated for any vaccine (YES/NO) –

By definition no MCV1 negative child can be MCV2 positive and therefore including them in the analysis biases all estimates. The entire analysis should be redone after excluding these MCV1 negative children.

Response: Thank you very much for your insightful comment. We agree with your observation that, by definition, a child who did not receive MCV1 cannot receive MCV2.

In our dataset, all children who were marked as MCV1-negative were also MCV2-negative. Therefore, no MCV1-negative and MCV2-positive cases were included in the analysis. We have now clarified this point in the manuscript under table -2 by stating

“All children recorded as MCV1-negative were also MCV2-negative; thus, no inconsistencies in the MCV1–MCV2 sequence were present in the dataset” to avoid any misinterpretation.

d.T5: Rows comparing Child received any vaccination vs. MCV2 status actually shows that a child not receiving any vaccine had higher odds (28/11) of getting MCV2 than those with any vaccine (239/348). The AOR would therefore be the inverse of 3.04.

However, as stated above including MCV0 children in prior vaccination status in the sample is erroneous as they can never become MCV2. The analysis should be redone.

Response: Thank you for your valuable comment. We fully agree with your observation. The issue you noted was due to a typographical error that occurred when transferring the multivariable table output. Specifically, the categories for MCV2 vaccination status were inadvertently reversed: MCV2 vaccinated was labeled as not vaccinated, and vice versa. To clarify, based on the correct cross-tabulation from our dataset, children who did not receive any vaccine had a lower odds of receiving MCV2 (11/28 = 0.393) compared to those who had received at least one vaccine (348/239 = 1.456). The correct cross tab output table is provided below for reference and we correct in revised manuscript (see Table -5).

Does your child had vaccinated for any vaccine? * 604. MCV2 Crosstabulation Count

604. MCV2Total NoYes

303.Does your child had vaccinated for any vaccine?Yes239348587 No 281139 Total267359626

4: DISCUSSION:

5.1 L265: Please mention country of GONDAR. Please rewrite the discussion section in light of revised analysis.

Response: Thank you for your comments we correct accordingly (see Discussion section Line 270, 284,301) .

**Do you want your identity to be public for this peer review?** For information about this choice, including consent withdrawal, please see our Privacy Policy

Reviewer #1: **Yes:** Mikhail P. Kostinov

Reviewer #3: No

---

## [Author Response · Author response to Decision Letter 2]

2 Sep 2025

Response to Reviewers

Title: Second-dose measles vaccination coverage and associated factors among children aged 19 to 35 months in Debre Birhan city, Ethiopia, 2024: a community-based cross-sectional study: PONE-D-24-57768R1

Reviewer #3:

1. Abstract: Please specify that the adjusted odds ratios were those associated with having received the second dose of measles vaccine. Significant factors associated with the second measles vaccine dose were …. Need a little more work to clarify language in the results section of the abstract

Response: Thank you for your insightful comment and suggestions, we correct it as Significant factors associated with the second dose of measles vaccination were…….

2. Introduction: this describes the issue clearly and presents an excellent rationale for this research.

Response: Thank you for acknowledging our rationale, which clearly reflects the motivation that led us to conduct this research.

3. Methods: Excellent description of study setting and the study sampling methodology is very sound and clearly described.

Response: We sincerely appreciate your positive evaluation of our description of the study setting and the methodological rigor of our sampling approach.

Specific comments:

1. If vaccination was not recorded on the infant immunization card or the health card, the parent was asked to recall whether measles two vaccination had been given or not (Line 144)

1.1 Can such parental recall be justified? For what proportion was this the method used to determine receipt of vaccine.

Response: Thank you for raising this important point. In our study, vaccination status was primarily verified using the infant immunization card or health card. However, in cases where these records were not available, parental recall was used as an alternative source of information. Parental recall has been commonly employed in similar population-based surveys, including the Demographic and Health Surveys (DHS), and has been shown to provide reasonably reliable estimates in the absence of written records. Even though this we acknowledge it in limitation section. In our study, parental recall accounted for -12/626=1.917 % of responses, we have now clarified this in the Methods section, Line 145. “In this study, parental recall was used for 12 out of 626 children (1.9%).

1.2 Was there a way that vaccination status was verified with the concerned health authorities?

Response: Thank you for this insightful comment. In our study, vaccination status was assessed using immunization/health cards and, when unavailable, caregiver recall. Direct verification with health facility or health authority records was not conducted, as our study was community-based and designed to capture information at the household level, similar to the methodology used in national Demographic and Health Surveys (DHS) and WHO immunization coverage surveys. We accept it, we put as limitation of this study and we state under limitation of the study as facility-level verification could have further strengthened the accuracy of vaccination status, see line 334-339.

Results

1. Good clear description of socio-demographic characteristics of mother-child pairs. Table, change ‘Categoties’ to ‘Categories’

Response: Thank you for your positive feedback on the description of the socio-demographic characteristics of mother–child pairs. We have corrected the typo in the table header, changing “Categoties” to “Categories.”

2. Table 2: Move the variables reporting parity and birth order and where child living to Table 1

Response: Thanks for your comment we move it to Table 1

3. Figure 2 legend: change ‘Figure 2: Measles two overage among children’ to’ Figure 2: Measles two coverage among children’

Response: thanks, we correct the error

4. Table 3 change ‘Ever has been post-pond or cancelled for vaccination’ to ‘Ever had a vaccination postponed or cancelled’

Response: thank you for your comment and we change it to “Ever had a vaccination postponed or cancelled”.

5. Change ‘Among these, five factors remained significant in the multivariable logistic regression analysis’ to ‘Among these, five factors remained independently associated with having received MCV2 in the multivariable logistic regression analysis’

Response: Thank you for your comment we correct it as ‘Among these, five factors remained independently associated with having received MCV2 in the multivariable logistic regression analysis….’

6. Change ‘Those children whose father’s educational status was primary, secondary, and college & above increased their odds of vaccinating’ to ‘In comparison to children whose fathers who had not received formal education, children whose father’s educational status was primary, secondary, and college & above had an increased odds of receiving MCV2’

Response: Thanks, we correct it accordingly, see line 248 to 250

7. Change ‘Mothers who had no schedules cancelled/postpone increased their odds of vaccinating their children for MCV2 by five times compared to mothers who had schedules cancelled/postpone (AOR: 4.9 (95% CI: 2.62, 9.32)).’ to ‘In comparison to children who had a vaccination postponed or cancelled, children who had not had a previous vaccination postponed or cancelled had an increased odds of receiving MCV2 (95% CI: 2.62, 9.32)).’

Response: Thank you, we correct it, see line 251 to 254

8. Change ‘In comparison to children whose mothers had low awareness regarding measles vaccination, children whose mothers had high awareness had an increased odds of receiving MCV2 (AOR:3.3 (95% CI: 2.12, 257 5.17)).’

Response: Thank you we correct it as you stated above, see line 254 to 256

9. Change ‘Similarly, the odds of vaccinating their children for MCV2 were 5 times (AOR: 2.62, 9.32) higher among children of mothers who had a good perception of measles vaccination compared to mothers with a poor perception (Table 5).’ to ‘In comparison to the children of mothers with a poor perception of measles vaccination, the children of mothers with a good perception of measles vaccination had an increased odds of receiving MCV2 (AOR: 2.62, 9.32)’.

Response: Thank you more, we also correct it as you stated above, see line 256 to 258

Discussion

1. Change ‘However, it is lower than the rates recorded Gondar in Ethiopia’ to ‘However, it is lower than the rates recorded in Gondar, Ethiopia’

1. Response: Thank for your critical insight, we correct as ‘However, it is lower than the rates recorded in Gondar, Ethiopia.’ See line 269

2. Start a new paragraph at ‘The disparities noted, especially in China and Japan,’ and change text to ‘The disparities noted, especially in comparison to China and Japan,’

Response: Thanks, corrected, as your recommendation

3. Start a new paragraph at ‘In contrast, the coverage reported in this study’

Response: Thank you, we start a new paragraph at ‘In contrast, the coverage reported in this study…’ See line 276

Limitations of study:

1. Can the number and % of children who did not have vaccination cards be described. The authors note that relying on parental recall may introduce bias which is true. To estimate the size of what effect of this bias we need to know for how many children the vaccination card was not available.

Response: Thank you, we describe it in % now under the methods section, see line 145

Figures

1. Figure 2 is unnecessary. This data can be presented in one sentence.

Response: Thank you for your comment. We included Figure 2 because presenting the overall coverage of MCV2 was one of our study objectives, and we felt that the figure would provide a visual emphasis. While the coverage is also summarized in the text “The overall coverage of MCV2 among children aged 19–35 months in Debre Berhan city was 57.3% (95% CI: 53.5–61.2)”, we believe the figure enhances reader understanding. Nevertheless, we are willing to remove it if the you still consider it unnecessary.

2. Figure 3. Correct the spelling errors in the title

Response: Thank you for your comment, I correct it accordingly.

---

## [Decision Letter · Decision Letter 2]

26 Dec 2025

Dear Dr. Moltot,

Thank you for submitting your manuscript to PLOS ONE. After careful consideration, we feel that it has merit but does not fully meet PLOS ONE’s publication criteria as it currently stands. Therefore, we invite you to submit a revised version of the manuscript that addresses the points raised during the review process.

We look forward to receiving your revised manuscript.

Kind regards,

Tebelay Dilnessa, MSc

Academic Editor

PLOS One

Journal Requirements:

Additional Editor Comments:

Line 30 and 31: The objective should be ‘SMART’.Line 93: Materials and methodsLine 125: The table legend should be explanatory in terms of person, place and time.The’ Authors’ contributions’ should be similar to the one generated during the first submission of the manuscript.The reference should be written based on Vancouver style of referencing together with PLOS ONE guideline for manuscript preparation.References 1-10 should be standardized.

Reviewers' comments:

Reviewer's Responses to Questions

**Comments to the Author**

Reviewer #1: (No Response)

Reviewer #4: All comments have been addressed

Reviewer #5: (No Response)

2. Is the manuscript technically sound, and do the data support the conclusions?

Reviewer #1: (No Response)

Reviewer #4: Yes

Reviewer #5: Partly

3. Has the statistical analysis been performed appropriately and rigorously?

Reviewer #1: (No Response)

Reviewer #4: No

Reviewer #5: Yes

4. Have the authors made all data underlying the findings in their manuscript fully available?

Reviewer #1: (No Response)

Reviewer #4: Yes

Reviewer #5: Yes

5. Is the manuscript presented in an intelligible fashion and written in standard English?

Reviewer #1: (No Response)

Reviewer #4: Yes

Reviewer #5: No

Reviewer #1: (No Response)

Reviewer #4: The manuscript addresses a relevant points and public health issue, but the title is repeatedly reported in Ethiopia. I have reviewed this cross-sectional study and provided my suggestions and comments, and the detailed suggestions and Comments are attached below.

Abstract

• The abstract needs grammatical, punctuation

• How you understand the herd immunity in Debre Birhan? And how do measure herd immunity?

• Significant factors associated with the second dose of measles vaccination were; father’s education level: primary (AOR: 1.63; 95% CI: 43 1.17–3.69), secondary (AOR: 2.04; 95% CI: 1.83–7.01), and college or above (AOR: 3.2; 95% 44 CI: 1.5–6.95). how do you interpret father’s education increases if father’s education influence measles vaccine coverage?

• Pease report or rewrite the variables level of awareness, and perception (like high, low medium) other wise it is difficult to interpret?

Introduction

In late 2023 and early 2024, Debre Birhan City, Ethiopia, experienced two measles outbreaks, despite high reported MCV1 vaccination coverage of 90–95% among children aged 19–35 months.

• Why this outbreak happened if the coverage due to this high vaccination coverage?

• Why do you select this child aged 19–35 group why no 15?

Method and Materials

• In line number 95-6 rewrite as Amhara National Regional State

• In line no 96-101 where you did obtain this information?

• A stratified sampling method, followed by systematic random sampling, was employed to select the study participants. Why stratified?

• How did you select the kebeles? write the name of the kebeles and their respective sample size?

• In line no 118-123 rewrite again please?

Results

• Line no 196-8 rewrite and remove the yielding replace it with?

• Please replace average age in mean age

• TV write the full name of this abbreviation

• Table 5 that shows the factors associated with MCV2 coverage among children aged 19–35 months, please add one column that indicates the CoR its CI and P-value.

• Please observe each cells carefully

• Revise each sentences in the result section line by line

Discussions

• Discussion section could be more elaborated compare them with existing literature, discuss implications

Conclusion

• Conclusions must align with the research questions, be directly grounded in the

study results, and recommendations should be directly based on the findings. Rewrite again please

Reviewer #5: Title: Why you select only second dose vaccine coverage:

Abstract:

• Line 33: “May 10 to 30, 2024” → “May 10–30, 2024” (use en dash).

• Ensuring consistency in describing the participants as children (not caregivers) and using precise language for the age range and sampling scope, I have assumed "in all over the paper" means applying this uniformly throughout the manuscript for clarity as a systematic random sampling method was used to recruit 626 children aged 19-35 months.

• Line 43: “Significant factors associated with the second dose of measles vaccination were;” → “Significant factors associated with second-dose measles vaccination coverage were:”

• Significant factors associated with receipt of the second dose of measles vaccination included father's education level (primary, secondary, or college). However, as expected, these sociodemographic variables are known to promote higher vaccination coverage in educated families. This is a well-documented and almost universally true finding in global health literature. In other words, higher parental education is generally associated with greater health literacy, better access to health information, and increased ability to navigate healthcare systems." It is not a novel discovery. Reporting it without deeper context or specific relevance to the local setting can make the study seem unoriginal and its conclusions overly simplistic.

Possible recommendation to improve it: To strengthen the impact of your study, contextualize the finding locally by explaining why father’s education specifically matters in Debre Birhan-such as the high proportion of male-headed households and cultural decision-making dynamics-and highlight what is new in your setting: emphasize that maternal education was not a significant predictor, unlike in many other studies, which underscores the unique and pivotal role of fathers in this community. Use stronger analytical language by framing the education effect as a clear “dose-response relationship,” where higher paternal education corresponds to progressively higher odds of vaccination.

Then, make actionable, father-focused recommendations, moving beyond generic awareness-raising to propose targeted strategies such as workplace outreach, male-tailored health messaging, and involving fathers directly in immunization visits. Finally, reframe the discussion to present father’s education not merely as a well-known sociodemographic factor, but as a key lever for designing locally effective, gender-sensitive interventions in Debre Birhan that can meaningfully improve second-dose measles vaccine coverage

• Line 49: “Therefore, to improve coverage, it is essential to strengthen health education, streamline vaccination schedules, and enhancing communication with parents are essential to boost coverage.” → This is repetitive. Simplify to: “Therefore, to improve coverage, it is essential to strengthen health education, streamline vaccination schedules, and enhance communication with parents.”

• To fully address measles prevention, I am strongly recommended to assess vaccine effectiveness in addition to coverage. This will help determine whether waning immunity or primary vaccine failure, key challenges in current measles control, are contributing to ongoing outbreaks.

• Other significant predictors included whether the child had received any prior vaccine, whether any vaccination schedules had been canceled or postponed, as well as caregivers’ levels of awareness and perception regarding measles vaccination. All of these are established factors known to influence vaccination uptake

• Why you include those who canceled or postponed vaccination schedules as study participants.

• How the one who took another vaccine before increases the vaccination coverage of measle vaccine?

Introduction:

• Well-written and clear. Good context and rationale.

• Line 90–93: Sentence: “This also raise concerns…” → “This raises concerns…”

• Ensure consistency in referencing: e.g., “[6, 8]” in revised manuscript (line 75) is fine.

Methods:

• Study Setting: Well described.

• Sample Size Formula (Line 115): “P = 0.48%” should be “P = 0.48” (without % symbol). Also, add a space after “0.48”.

• Sampling: Clear and reproducible.

• Line 145: “In this study, parental recall was used for 12 out of 626 children (1.9%).” → Good addition.

• Operational Definitions: Clear, but ensure Awareness and Perception scoring is explicitly referenced (mean/median).

• Ethics: Well documented.

Results:

• Tables: Ensure all table titles are consistent (e.g., Table 1, 2, 3, 5).

• Figure 2 Title: “Measles two average” → “Measles second-dose coverage”

• Line 225–227: The coverage is stated clearly. Figure 2 is simple but acceptable.

• Table 5: Ensure column alignment. “Not vaccinate” → “Not vaccinated”.

• AOR Reporting: In text (Lines 248–258), ensure confidence intervals are formatted consistently: “(95% CI: 1.17–3.69)”.

Discussion:

• Line 283–284: “Gondar, Ethiopia (75.68%)” → Consider citing the correct reference.

• Line 285–286: “The disparities noted, especially in China and JapanThe disparities noted, especially in China and Japan,” → Duplicate text. Remove repetition.

• Line 301–305: Well articulated.

• Line 332–333: “Ethiopia; Gondar,” → “Ethiopia, including Gondar,”

• Line 339–347: The paragraph on strategies is relevant but could be more concise. Consider integrating into conclusion.

Limitations:

• Clearly stated. You may add: “Recall bias was minimal given only 1.9% relied on recall.”

Conclusion and Recommendations:

• Clear and practical.

• Line 358–365: Repetitive with abstract. Ensure concise, actionable recommendations.

• Suggest specifying: “Targeted health education for fathers, reminder systems for appointments, and community engagement.”

References:

• Formatting is inconsistent (some with full URLs, some without). Follow PLOS ONE citation style.

• Ensure all hyperlinks are properly bracketed and accessible.

• Example: Reference 1“Measles cases surge worldwide…” URL is very long; consider shortening or using DOI if available.

• Check for duplicate or misplaced references (e.g., references 6 and 8 are same?).

General Comments:

• Language: Mostly clear but some long sentences. Break them for readability.

• Typos: “overage” → “coverage” in Figure 2.

• Consistency: Use “MCV2” throughout instead of “measles two”.

• Figures/Tables: Ensure all are cited in text and legends are clear.

• Response to Reviewers: Well addressed. Ensure all changes are reflected in the main manuscript (some tracked changes remain).

Recommended Actions Before Submission:

1. Run a spell-check and grammar check.

2. Standardize reference formatting.

3. Verify all numbers in tables match text.

4. Consider removing Figure 2 if space is limited

**Do you want your identity to be public for this peer review?** For information about this choice, including consent withdrawal, please see our Privacy Policy

Reviewer #1: **Yes:** Mikhail P. Kostinov

Reviewer #4: **Yes:** Habtamu Belew

Reviewer #5: No

---

## [Author Response · Author response to Decision Letter 3]

23 Jan 2026

Reviewers' comments:

Reviewer #4:

Abstract

• The abstract needs grammatical, punctuation

Response: Thank you for your comment. We have incorporated this suggestion, and it has strengthened the abstract.

• How you understand the herd immunity in Debre Birhan? And how do measure herd immunity?

Response: We thank the reviewer for the opportunity to clarify our methodology.

We understand herd immunity as the critical coverage (Cc) needed to reduce the effective reproduction number (Re) below 1, calculated as Cc=1−(1/R0). For measles, with an R0 of ~15, the herd immunity threshold is approximately 93-94%. Accounting for two-dose vaccine efficacy (~97%), the required two-dose coverage target becomes ~95%.

• Significant factors associated with the second dose of measles vaccination were; father’s education level: primary (AOR: 1.63; 95% CI: 43 1.17–3.69), secondary (AOR: 2.04; 95% CI: 1.83–7.01), and college or above (AOR: 3.2; 95% 44 CI: 1.5–6.95). how do you interpret father’s education increases if father’s education influence measles vaccine coverage?

Response: We thank the reviewer for this question, which allows us to clarify an important detail omitted from the abstract.

Interpretation: In our multivariable logistic regression model, the reference category for father's education was "no formal education." Therefore, the reported Adjusted Odds Ratios (AORs) are interpreted relative to this baseline.

• AOR = 1.63 for primary education means that children whose fathers have a primary education have 1.63 times higher odds of receiving the second measles dose compared to children whose fathers have no formal education.

• AOR = 2.04 for secondary education indicates 2.04 times higher odds compared to the "no education" group.

• AOR = 3.2 for college or above indicates 3.2 times higher odds compared to the "no education" group.

Conclusion: The results clearly demonstrate a positive gradient—higher paternal education is significantly associated with increased odds of a child completing the second measles vaccine dose. This aligns with the global evidence that parental education, often a proxy for health literacy and access to information, is a strong enabler of childhood vaccination.

• Pease report or rewrite the variables level of awareness, and perception (like high, low medium) otherwise it is difficult to interpret?

Response: Thank you for your comments, we correct it.

Introduction

In late 2023 and early 2024, Debre Birhan City, Ethiopia, experienced two measles outbreaks, despite high reported MCV1 vaccination coverage of 90–95% among children aged 19–35 months.

• Why this outbreak happened if the coverage due to this high vaccination coverage?

Response: We thank the reviewer for this vital question, which directly addresses the central paradox motivating our research. The occurrence of outbreaks despite high MCV1 coverage can be explained by several interconnected factors, and our study provides empirical evidence for the most critical one in Debre Birhan.

The core explanation lies in the difference between MCV1 and MCV2 coverage and the nature of measles immunity:

1. High MCV1 coverage ≠ Herd Immunity: While MCV1 coverage of 90-95% is commendable, it is insufficient for measles eradication. Herd immunity for measles requires very high population immunity (typically ≥94%) with a two-dose regimen. The first dose has an efficacy of about 93%. Even with 95% MCV1 coverage, this leaves a significant immune gap (approximately 30-35% of the birth cohort may still be susceptible when accounting for vaccine efficacy). This susceptible pool is sufficient to sustain outbreaks, especially in densely populated urban areas like Debre Birhan.

2. Our Study's Critical Finding: Our research identified that second-dose (MCV2) coverage in the same age group is only 57.3%. This represents a catastrophic "coverage gap" of over 30 percentage points between dose one and dose two. This means a large proportion of children who received the first dose did not receive the crucial second dose needed to boost immunity to near 100%.

• Why do you select this child aged 19–35 group why no 15?

Response: We thank the reviewer for raising this important methodological point. The Ethiopian national immunization schedule recommends the second dose of measles-containing vaccine (MCV2) at 15 months of age. We selected the age range of 19–35 months to specifically identify children who were true defaulters—those who had passed the recommended age by at least 4 months. This allowed us to accurately assess routine MCV2 coverage and its barriers, rather than including children who were still within the standard grace period for the 15-month vaccination.

Method and Materials

• In line number 95-6 rewrite as Amhara National Regional State

Response: Thank you we correct it

• In line no 96-101 where you did obtain this information?

Response: Thanks for your comment I obtain this information from ‘Population-of-Zones-and-Weredas-Projected-as-of-July-2023, by Ethiopian statistical service

• A stratified sampling method, followed by systematic random sampling, was employed to select the study participants. Why stratified?

Response: Thanks for your interesting comment, stratified sampling was used at the first stage because the five sub-cities from which we sampled were not homogeneous. They contained distinct socio-demographic and geographic strata, most notably that some sub-cities included predominantly rural kebeles while others were composed entirely of urban kebeles.

To ensure our sample was representative of both these key settings (urban and rural), we first stratified the population by sub-city. This allowed us to guarantee proportional representation of these different community types before proceeding with random selection within the chosen strata.

• How did you select the kebeles? write the name of the kebeles and their respective sample size?

Response: Thank you for this for requesting clarification. The eight selected kebeles and their respective sample sizes are detailed under figure one. As stated in the manuscript, four kebeles were selected from each of the two chosen sub-cities.

Selected Kebeles and Sample Sizes:

From Etege Taytu Sub-City:

o Giorgis Kebele: 81 participants

o Ayer Tena Kebele: 82 participants

o Bakelo Kebele: 83 participants

o Saria Kebele: 82 participants

From Tebasse Sub-City:

o Geberal Kebele: 77 participants

o Bahir Hali Kebele: 76 participants

o Addis Genet Kebele: 75 participants

o Mesalemya Kebele: 78 participants

Total Sample Size (N): 634 participants

• In line no 118-123 rewrite again please?

Response: Thank you and we rewrite it, see line 118-123

Results

• Line no 196-8 remove the yielding replace it with?

Response: Thank you for your comment and we replace ‘yielding’ by ‘with’

• Please replace average age in mean age

Response: Thanks, we replace ‘average’ by ‘mean’

• TV write the full name of this abbreviation

Response: Thank you we rewrite it with its full name

• Table 5 that shows the factors associated with MCV2 coverage among children aged 19–35 months, please add one column that indicates the CoR its CI.

Response: Thank you very much we removed it due to recommendation of one of the previous reviewers, but now we accept your recommendation and add again it

• Please observe each cell carefully

Response: Thank very much we recheck each cell of the table

• Revise each sentence in the result section line by line

Response: Thanks very much we revise the whole result section, we correct some inconvenient words.

Conclusion

• Conclusions must align with the research questions, be directly grounded in the

study results, and recommendations should be directly based on the findings. Rewrite again please

Response: Thank you for your valuable and constructive feedback. We rewrite it now.

Reviewer #5:

Title: Why you select only second dose vaccine coverage

Response: Thank you for your question. In late 2023 and early 2024, Debre Birhan City, Ethiopia, experienced two measles outbreaks, despite high reported MCV1 vaccination coverage of 90–95% among children aged 19–35 months, but thier is no clear image regardeding MCV2. This gaps, raise concerns about the coverage, and factors affecting MCV2 administration. Therefore we plan to conduct this research specifically on MCV2.

Abstract:

• Line 33: “May 10 to 30, 2024” → “May 10–30, 2024” (use en dash).

Response: Thank you for your comment and we make it -

• Ensuring consistency in describing the participants as children (not caregivers) and using precise language for the age range and sampling scope, I have assumed "in all over the paper" means applying this uniformly throughout the manuscript for clarity as a systematic random sampling method was used to recruit 626 children aged 19-35 months.

Response: Thank you very much for your comment, we revise and correct it in all over the paper.

• Line 43: “Significant factors associated with the second dose of measles vaccination were;” → “Significant factors associated with second-dose measles vaccination coverage were:”

Response: Thank you for your insightful recommendation and we correct it as you state above

• Significant factors associated with receipt of the second dose of measles vaccination included father's education level (primary, secondary, or college). However, as expected, these sociodemographic variables are known to promote higher vaccination coverage in educated families. This is a well-documented and almost universally true finding in global health literature. In other words, higher parental education is generally associated with greater health literacy, better access to health information, and increased ability to navigate healthcare systems." It is not a novel discovery. Reporting it without deeper context or specific relevance to the local setting can make the study seem unoriginal and its conclusions overly simplistic.

Possible recommendation to improve it: To strengthen the impact of your study, contextualize the finding locally by explaining why father’s education specifically matters in Debre Birhan-such as the high proportion of male-headed households and cultural decision-making dynamics-and highlight what is new in your setting: emphasize that maternal education was not a significant predictor, unlike in many other studies, which underscores the unique and pivotal role of fathers in this community. Use stronger analytical language by framing the education effect as a clear “dose-response relationship,” where higher paternal education corresponds to progressively higher odds of vaccination.

Then, make actionable, father-focused recommendations, moving beyond generic awareness-raising to propose targeted strategies such as workplace outreach, male-tailored health messaging, and involving fathers directly in immunization visits. Finally, reframe the discussion to present father’s education not merely as a well-known sociodemographic factor, but as a key lever for designing locally effective, gender-sensitive interventions in Debre Birhan that can meaningfully improve second-dose measles vaccine coverage

Response: Thank you very much, we try our best to improve the recommendation section of the Abstract

• Line 49: “Therefore, to improve coverage, it is essential to strengthen health education, streamline vaccination schedules, and enhancing communication with parents are essential to boost coverage.” → This is repetitive. Simplify to: “Therefore, to improve coverage, it is essential to strengthen health education, streamline vaccination schedules, and enhance communication with parents.”

Response: Thank you very much for your insightful suggestion and we correct it

• To fully address measles prevention, I am strongly recommended to assess vaccine effectiveness in addition to coverage. This will help determine whether waning immunity or primary vaccine failure, key challenges in current measles control, are contributing to ongoing outbreaks.

Response: Thank you very much, we will consider this for our future work

• Why you include those who canceled or postponed vaccination schedules as study participants.

Response: Thank you for this important methodological question. We included children with a history of canceled or postponed vaccination schedules as study participants for the following key reasons,

1. The target population for MCV2 in this study was all children aged 19-35 months residing in the city. Excluding children based on healthcare-seeking behaviors or prior interactions with the system (like cancellations) would create a selection bias, artificially inflating coverage estimates by only including children more likely to be reached or from more engaged families.

2. Capturing "Missed Opportunities" and System Barriers: A history of cancellation or postponement is itself a critical factor associated with ultimately missing the MCV2 dose. By including these children, we could quantify the impact of this specific barrier on final coverage.

In general, this approach was taken to ensure a true measurement of population-level coverage and to allow for the analysis of missed opportunities and system-related barriers as potential factors associated with incomplete vaccination.

• How the one who took another vaccine before increases the vaccination coverage of measle vaccine?

Response: A child who has received any prior vaccine (e.g., Pentavalent, PCV, Vitamin A) before the measles-containing vaccine (MCV) schedule is more likely to receive MCV2 due to several interrelated factors:

1. Indicator of successful health system contact: Receiving a prior vaccine signifies that the child is enrolled in the immunization system, the caregiver knows the vaccination site/schedule, and at least one successful contact with the health service has occurred. This establishes a pathway for subsequent vaccinations.

2. Reflection of caregiver health-seeking behavior: Caregivers who proactively complete early vaccines generally exhibit higher health literacy, stronger belief in vaccine benefits, and greater commitment to the recommended schedule. This positive behavior pattern continues, increasing the likelihood they will return for MCV2.

Introduction:

• Well-written and clear. Good context and rationale.

Response: Thank you very much

• Line 90–93: Sentence: “This also raise concerns…” → “This raises concerns…”

Response: Thank you very much and we correct it

• Ensure consistency in referencing: e.g., “[6, 8]” in revised manuscript (line 75) is fine.

Response: Than you we revise it

Methods:

• Study Setting: Well described.

Response: Thank you very much

• Sample Size Formula (Line 115): “P = 0.48%” should be “P = 0.48” (without % symbol). Also, add a space after “0.48”.

Response: Thank you for your critical review and we delete %

• Sampling: Clear and reproducible.

Response: Thank you very much

• Line 145: “In this study, parental recall was used for 12 out of 626 children (1.9%).” → Good addition.

Response: Thank you very much

• Operational Definitions: Clear, but ensure Awareness and Perception scoring is explicitly referenced (mean/median).

Response: Thank you very much, we ensure it and we use mean for awareness and median for perception

• Ethics: Well documented.

Response: Thank you very much

Results:

• Tables: Ensure all table titles are consistent (e.g., Table 1, 2, 3, 5).

Response: Thank you for your comment, we make it consistent

• Figure 2 Title: “Measles two average” → “Measles second-dose coverage”

Response: Thank you, we correct it as recommended

• Line 225–227: The coverage is stated clearly. Figure 2 is simple but acceptable.

Response: Thank you we left it as it is

• Table 5: Ensure column alignment. “Not vaccinate” → “Not vaccinated”.

Response: Thanks for your critical observation and we correct it.

• AOR Reporting: In text (Lines 248–258), ensure confidence intervals are formatted consistently: “(95% CI: 1.17–3.69)”.

Response: Thank you we make it consistent

Discussion:

• Line 283–284: “Gondar, Ethiopia (75.68%)” → Consider citing the correct reference.

Response: Thank you we recheck it and the citation is corrected

• Line 285–286: “The disparities noted, especially in China and Japan The disparities noted, especially in China and Japan

---

## [Editor Report · Decision Letter 3]

26 Jan 2026

Second-dose measles vaccination coverage and associated factors among children aged 19 to 35 months in Debre Birhan city, Ethiopia, 2024: a community-based cross-sectional study

PONE-D-24-57768R3

Dear Dr. Moltot,

We’re pleased to inform you that your manuscript has been judged scientifically suitable for publication and will be formally accepted for publication once it meets all outstanding technical requirements.

Kind regards,

Tebelay Dilnessa, MSc

Academic Editor

PLOS One

Additional Editor Comments (optional):

Line 89: This study aimed to assess MCV2 coverage and…………Line 92: **Materials and methods**
---

## [Editor Report · Acceptance letter]

PONE-D-24-57768R3

PLOS One

Dear Dr. Moltot,

I'm pleased to inform you that your manuscript has been deemed suitable for publication in PLOS One. Congratulations! Your manuscript is now being handed over to our production team.

Kind regards,

on behalf of

Dr. Tebelay Dilnessa

Academic Editor

PLOS One